# PredIL13: Stacking a variety of machine and deep learning methods with ESM-2 language model for identifying IL13-inducing peptides

**Hiroyuki Kurata** ⓘ*, **Md. Harun-Or-Roshid** ⓘ, **Sho Tsukiyama, Kazuhiro Maeda**

Department of Bioscience and Bioinformatics, Kyushu Institute of Technology, Kawazu, Iizuka, Fukuoka, Japan

* kurata@bio.kyutech.ac.jp

**Data Availability Statement:** The source codes are freely accessible at https://github.com/kuratahiroyuki/PredIL13. The web application is

## Abstract

Interleukin (IL)-13 has emerged as one of the recently identified cytokine. Since IL-13 causes the severity of COVID-19 and alters crucial biological processes, it is urgent to explore novel molecules or peptides capable of including IL-13. Computational prediction has received attention as a complementary method to in-vivo and in-vitro experimental identification of IL-13 inducing peptides, because experimental identification is time-consuming, laborious, and expensive. A few computational tools have been presented, including the IL13Pred and iIL13Pred. To increase prediction capability, we have developed PredIL13, a cutting-edge ensemble learning method with the latest ESM-2 protein language model. This method stacked the probability scores outputted by 168 single-feature machine/deep learning models, and then trained a logistic regression-based meta-classifier with the stacked probability score vectors. The key technology was to implement ESM-2 and to select the optimal single-feature models according to their absolute weight coefficient for logistic regression (AWCLR), an indicator of the importance of each single-feature model. Especially, the sequential deletion of single-feature models based on the iterative AWCLR ranking (SDIWC) method constructed the meta-classifier consisting of the top 16 single-feature models, named PredIL13, while considering the model's accuracy. The PredIL13 greatly outperformed the-state-of-the-art predictors, thus is an invaluable tool for accelerating the detection of IL13-inducing peptide within the human genome.

## Introduction

A cascade of cytokines, marked by the overproduction of inflammatory signaling molecules such as Interleukin (IL)-1, IL-2, IL-6, IL-13, IL-17, Interferon-γ, and TNF-α is identified as a physiological and pathological factor intricately linked to the severity of the Coronavirus disease 2019 (COVID-19) [1, 2]. In-vitro experimental investigations, which were supported by in-vivo data and enriched by insights resulting from single-cell RNA-sequencing, revealed that inflammatory agents present in the blood serum of COVID-19 patients trigger dysfunction in the endothelial cells. Such dysfunction is closely connected to COVID-19-related

freely available at http://kurata35.bio.kyutech.ac.jp/PredIL13.

**Funding:** This work is supported by Japan Society for the Promotion of Science(JSPS) with grant number 22H03688. In relation to this, the funder had no role in study design, data collection and analysis, decision to publish, or preparation of the manuscript. There was no additional external funding received for this study.

**Competing interests:** The authors have declared that no competing interests exist.

endotheliopathy, providing some evidences of the detrimental functions caused by inflammatory cytokines [3].

IL-13 has emerged as one of the recently identified cytokines as contributors to the severity of COVID-19 [4, 5]. IL-13, a versatile cytokine, is discharged by T-Helper 2 (Th-2) cells, basophils, mast cells, eosinophils, and natural killer cells. Analogous to IL-4, this cytokine plays a crucial role in Th-2-mediated immunity, encompassing responses to allergic reactions and parasitic infections [6]. Actually, IL-13 triggers the transition to IgG4 and IgE antibodies in naive human B cells [7] and proves indispensable in expelling gastrointestinal nematodes [8]. It stands out as a pivotal mediator in the airway inflammation observed in conditions such as asthma and reactive airway diseases [9].

Since IL-13 causes the severity of COVID-19 and alters crucial biological processes, it is urgent to explore novel molecules capable of modulating IL-13. Computational or in-silico prediction has received attention as a complementary method to in-vivo and in-vitro experimental identification of IL-13 [10]. To data only a few predictors have been presented and the development of IL-13 predictors has just begun. Jain et al. introduced the first predictor of IL13Pred in 2022, which was designed to categorize peptides into those inducing IL-13 and those lacking this property [11]. They presented a benchmark dataset that comprised 313 experimentally validated IL-13-inducing peptides retrieved from the immune epitope database [12]. In addition, 2908 non-IL-13-inducing peptides were extracted from the same database as negative datasets. They used the Pfeature algorithm to compute 9151 features for each peptide, executed feature selection using the linear support vector classifier with the L1 penalty method, resulting in the identification of 95 relevant features. They employed a decision tree-based algorithm to predict IL-13-inducing peptides. Arora et al. developed iIL13Pred that used seven conventional machine learning (ML) classifiers: decision tree, Gaussian Naïve Bayes, k-Nearest Neighbour (KN), Logistic Regression (LR), Support Vector Machine (SVM), Random Forest (RF), and eXtreme Gradient Boosting (XGB) while introducing a multivariate feature selection approach [13].

In this study, we have developed PredIL13, a cutting-edge ensemble learning model that accurately identified IL-13 inducing peptides, as shown in Fig 1. This method stacked the probability scores generated by168 single-feature machine/deep learning models, then trained a logistic regression-based meta-classifier with the stacked probability score vectors. The key technology was to implement Evolutionary Scale Modeling-2 (ESM-2) [14] language model and to select the optimal single-feature models according to their absolute weight coefficient for logistic regression (AWCLR), an indicator of the importance of each single-feature model. Especially, the sequential deletion of single-feature models based on the iterative AWCLR ranking (SDIWC) method constructed the meta-classifier with the top 16 single-feature models, named IL13Pred, while considering the prediction accuracy. The SDIWC method enables us to intelligibly determine the optimal number of single-feature models. The PredIL13 greatly outperformed the-state-of-the-art predictors, thus is an invaluable tool in accelerating the detection of IL13-inducing peptide within the human genome. To aid the scientific community in identifying latent IL-13 inducing peptides, we present a web application and standalone programs of the proposed predictor, which can be freely accessed. The web application is freely available at http://kurata35.bio.kyutech.ac.jp/PredIL13. The source codes are freely accessible at https://github.com/kuratahiroyuki/PredIL13.

## Materials and methods

### Overall framework of PredIL13

Fig 1 provides an overview of the process employed in the development of the IL13- predictor. This multifaceted procedure consists of the following steps: dataset preparation, construction

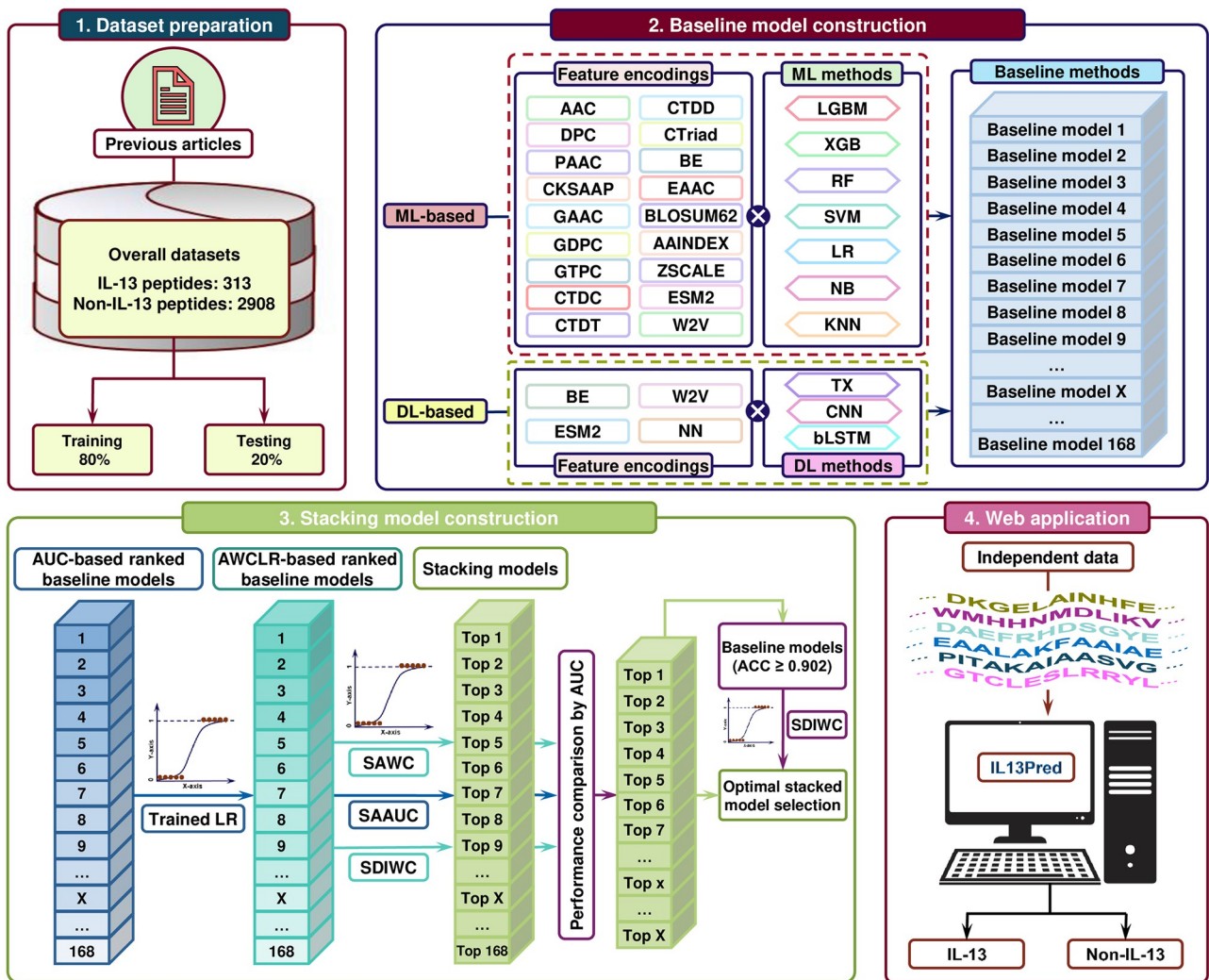

**Fig 1. An overview of the process employed in the development of the PredIL13 predictor.** This process comprises the following steps: preparing the dataset, extracting features, training the model, evaluating its performance, and creating web application and standalone programs.

of single-feature models using ML and DL methods, construction of the meta-classifier that stacks the single-feature models, evaluation, and web application/standalone program construction.

## Dataset preparation

We used the dataset used in the latest study [11, 13]. For the sake of comparison, all the datasets including the positive and negative datasets used in this study were obtained from the original study [11]. The positive dataset included 313 IL-13-inducing peptides whereas the negative dataset included 2908 non-IL-13-inducing peptides. The entire dataset was randomly divided into the training and test datasets at a ratio of 4 to 1, respectively. The training dataset was used for 5-fold cross validation (CV), where the ratio of the training and validation datasets was 4 to 1. While the ratio of positive to negative samples in the dataset was imbalanced, we neither used over-sampling nor under-sampling methods.

## Feature encoding method

The amino acid sequences with a different length were encoded using 21 different encoding techniques. These methods included Amino Acid Composition (AAC), Di-Peptide Composition (DPC), Composition of k-spaced Amino Acid Pairs (CKSAAP), Grouped Amino Acid Composition (GAAC), Conjoint Triad (CTriad), Composition/Transition/Distribution (CTD) Composition (CTDC), CTD Transition (CTDT), CTD Distribution (CTDD), Binary Encoding (BE), Enhanced Amino Acid Composition (EAAC), Amino Acid indices (AAindex), BLOSUM62, Z-Scale (ZSCALE), Evolutionary Scale Modeling-2 (ESM-2), and Word2Vec (W2V) with different Kmers. All these sequence encoding techniques can be readily computed using the open-source software packages including iLearn [15]. These encoding techniques capture a variety of properties including compositional, position-order, evolutional and physicochemical characteristics, and linguistic patterns/distributions. Each of these encoding methodologies brings a unique lens to the sequence and contributes to a comprehensive and detailed analysis. Details in each encoding method are described below:

## Composition-based encoding

Generally, the Kmer ($k$) encodes an amino acid sequence ($\{R_i\}$ ($i = 1,2, \ldots, L$), $L$ is the length of a sequence) as the occurrence frequencies of consecutive amino acids of length $k$. The Kmer encoding provides $20^k$ features to each sequence, given by:

$$f(q_k) = \frac{f(q_k)}{L - k + 1} \tag{1}$$

where $q_k$ is any one of the consecutive Kmer amino acids and $f(AA_k)$ is the occurrence number of $q_k$ [15, 16]. For example, at Kmer = 2 a sequence is encoded as 400 ($20^2$) descriptors (features) representing the frequency of all the dinucleotides (2-mers). Amino Acid Composition (AAC) is a variant of Kmer = 1 that gives 20 features for a sequence. Di-Peptide acid Composition (DPC) and Tri-Peptide Composition (TPC) are the variants of Kmer = 2 that gives 400 ($20^2$) descriptors and Kmer = 3 that gives 8000 ($20^3$) features, respectively.

## Pseudo-Amino Acid Composition (PAAC)

PAAC encodes the amino acid sequences mainly using a matrix of amino-acid frequencies, which deals with proteins without significant sequential homology to other proteins [17]. Compared to AAC, PAAC has 25 descriptors that can include some local sequence order information or a series of rank-different correlation factors along a protein sequence.

## Composition of k-spaced Amino Acid Pairs (CKSAAP)

The CKSAAP feature encoding calculates the frequency of amino acid pairs separated by any $k$ residues ($k = 0, 1, 2, \ldots, 5$) [18]. The default maximum value of $k$ is 5. For example, it generates 400 descriptors that correspond to DPC at $k = 0$.

## Grouped Amino Acid Composition (GAAC)

The 20 amino acid types are categorized into five classes according to their physicochemical properties, including hydrophobicity, charge and molecular size [19]. The five classes include the aliphatic group (g1: GAVLMI), aromatic group (g2: FYW), positive charge group (g3: KRH), negative charged group (g4: DE) and uncharged group (g5: STCPNQ). GAAC encodes an amino acid sequence as the occurrence frequencies of the different grouped amino acids, generating 5 features. The 400 di-peptide types are categorized into 25 classes based on the

physicochemical properties in the same manner as GAAC. Grouped Di-Peptide Composition (GDPC) encodes an amino acid sequence as the occurrence frequencies with respect to the different grouped dipeptides, generating 25 features. The Grouped Tri-Peptide Composition (GTPC) encodes an amino acid sequence as the occurrence frequencies with the different grouped tripeptides, where 125 descriptors are used.

### Conjoint Triad (CTriad)

Twenty standard amino acids can be divided into 7 groups based on the dipoles and volumes of the side chains. The Conjoint Triad descriptor (CTriad) considers the properties of one amino acid and its vicinal amino acids by regarding any three continuous amino acids as a single group, generating a $7 \times 7 \times 7$-D feature vector for a sequence [20].

### Composition/Transition/Distribution (CTD)

The Composition, Transition and Distribution (CTD) features represent the amino acid distribution patterns of seven structural or physicochemical properties, including hydrophobicity, normalized Van der Waals Volume, polarity, polarizability, charge, secondary structures and solvent accessibility [21, 22]. The CTD descriptors are calculated by transforming the amino acid sequence into a vector of the specific structural or physicochemical properties of amino acid residues, where 20 amino acids are classified into three groups (polar, neutral and hydrophobic) for each of the seven different physicochemical features.

The CTD encodes an amino acid sequence as a combination of the three different matrices: Composition, Transition, and Distribution. We can obtain any of the three matrices by executing the corresponding encodings: CTDC, CTDT, and CTDD. The CTDC calculates the grouped amino acid composition for each property, and generates a 39D vector for each sequence, where the number of rows is the sequence length and the number of columns is the number of groups times the number of properties. The CTDT calculates the grouped amino acid transition for each property to generates a 39-D feature matrix. The CTDD calculates 15 values of the amino acids for each property. The Distribution descriptor consists of the five values for each of the three groups, defined as the corresponding fraction of the entire sequence, given by 0, 25, 50, 75 and 100% of occurrences. It generates a 195-D feature vector.

### Position-based encoding

**Binary Encoding (BE).**   BE converts a single amino acid into a 20-dimensional binary vector. For example, the amino acids A, C, and Y of are represented as (10000000000000000000), (01000000000000000000), and (00000000000000000001), respectively. Therefore, an amino acid sequence with a length of $L$ can be represented as a $20L$ dimensional feature vector.

**NN.**   NN encoding (nn.Embedding in PyTorch) converts a single amino acid into an index, which is further transformed into a fixed-size feature vectors by using a generated lookup table that refers embeddings in a fixed dictionary and size [23].

**EAAC.**   The Enhanced Amino Acid Composition (EAAC) feature calculates the AAC based on the sequence window of fixed length (the default value is 5) that continuously slides from the N- to C-terminus of each peptide [15]. EAAC encodes a sequence with length $L$ as $20 \times (L - k + 1)$ dimensional feature vector, given by:

$$\left(b_{i,1}, b_{i,2}, \ldots, b_{i,L-k+1}\right)$$

$$b_{ij} = \frac{\sum_{m=j}^{j+k-1} f(R_m)}{k}, \quad f(R_m) = \begin{cases} 1 & if \ R_m = q_i \\ 0 & others \end{cases} \tag{2}$$

where $R_m$ represents the $m$th amino acid in a sequence and $q_i \in \{A, R, N, D, C, Q, E, G, H, I, L, K, M, F, P, S, T, W, Y, V\}$.

## Amino acid index

Physicochemical properties of amino acids are the most intuitive features for representing biochemical reactions and have been extensively applied in bioinformatics research. The Amino Acid indices database (AAindex) collects many published indices representing physicochemical properties of amino acids [24]. Each physicochemical property has a set of 20 numerical value for all amino acids. Currently, 544 physicochemical properties can be retrieved from the AAindex database. After removing the physicochemical properties with NA (not available) for any of the amino acids, AAindex generates a vector of 531 mean values for each amino acid reside in a sequence.

## BLOSUM62

The BLOcks SUbstitution Matrix (BLOSUM) is a substitution matrix used for sequence alignment of proteins and scores the alignments between evolutionarily divergent protein sequences [25]. It scans the BLOCKS database for very conserved regions of protein families based on their similarity, counts the relative frequencies of amino acids, and calculate their substitution probabilities as a log-odds score for each of the 210 possible substitution pairs of the 20 standard amino acids. The BLOSUM62 is the matrix built using sequences with less than 62% similarity. The BLOSUM62 generates 20$L$-D feature vector to represent in a sequence, where each row in the BLOSUM62 matrix is adopted to encode one of 20 amino acids.

## Z-Scale (ZSCALE)

Z-Scale characterizes each amino acid in a sequence by five physicochemical descriptors [26] and generate 25$L$ D feature vector for each sequence. It improves the original Z-scales [27] by introducing two more Z-scales.

## Language model

ESM-2 is a transformer-based language model that uses an attention mechanism to learn interaction patterns between pairs of amino acids [14]. ESM-2 can leverage evolutionary information from diverse protein sequences, enabling accurate predictions of 3D structures. ESM-2 was trained with 15 billion parameters on the protein sequences from the UniRef database [28], where it is tasked with predicting the 15% masked amino acids using the remaining 85% amino acid sequences.

W2V is invented to obtain the distributed representation of words in the field of natural language processing [29]. In W2V, the weights in a neural network are determined by learning the context of words to provide the distributed representation that encodes different linguistic regularities and patterns. In this study the Kmer in amino acid sequences are regarded as a single word and each peptide sequence is represented by multiple consecutive Kmer amino acids. Here, Kmer amino acids and each peptide sequences correspond to words and sentences in natural language. We trained a Skip-gram-based W2V model in the SWISS-PROT database [30] to learn the appearance pattern of Kmer by using the genism of the python package [31]. This study used Kmer of 1, 2, 3, and 4, feature size of 128, epoch of 100, window size of 40, sg of 1. At Kmer of K, we named W2V_K.

## ML and DL classifiers

We employed seven different classifiers: Random Forest (RF) [32], Extreme Gradient Boosting (XGB) [33], LightGBM (LGBM) [34], Support Vector Machine (SVM) [35], K-nearest neighbor (KN), Naive Bayes (NB), and Logistic regression (LR). In addition, we used three DL methods: Transformer encoder network (TX), Convolutional Neural Network (CNN) and Bidirectional Long Short Term Memory (bLSTM) algorithm [36].

**RF.**   RF is an ensemble learning method that constructs numerous decision trees in the training phase. This approach employs bagging or bootstrap aggregating, generating multiple subsets of the original dataset randomly with replacement. A decision tree model is then established for each subset. During the prediction phase, RF combines the outputs of each tree to deliver a conclusive prediction. RF effectively addresses overfitting by reducing variance without augmenting bias, establishing itself as a robust tool for diverse applications.

**XGB.**   XGB stands out as a high-performance, adaptable, and easily transportable gradient boosting method. In a step-by-step fashion, XGB assembles an ensemble of weak prediction models, typically in the form of decision trees. Each subsequent tree rectifies the prediction errors of its predecessor. To counteract overfitting, XGB incorporates a regularization parameter, simplifying the model for enhanced robustness across a spectrum of predictive tasks.

**LGBM.**   LGBM stands as a gradient boosting platform harnessing tree-based learning algorithms, presenting heightened efficiency and swiftness compared to counterparts such as XGB and CatBoost. The distinctive features of LGBM encompass adept handling of expansive datasets, superior effectiveness, and rapid execution. What distinguishes it is the employment of a histogram-based algorithm, discretizing continuous feature values into distinct bins, a departure from conventional tree-based methods. This approach accelerates the training procedure while concurrently reducing memory consumption.

**SVM.**   SVM's fundamental idea involves placing each data point in an $N$-dimensional space ($N$ denotes the number of features) and defining a hyperplane that effectively separates the data points into distinct classes. The selected hyperplane seeks to maximize the margin between these classes. In cases where the data is not linearly separable, SVM employs a kernel trick to transform the input space into a higher dimension, enabling a hyperplane to delineate the data. The learning model constructs a boundary line that segregates data points into various classes. In binary classification, this decision boundary adopts the approach of creating the widest street, maximizing the distance to the closest data points from each class.

**LR.**   LR is used to predict the likelihood of categorical dependent variables, which makes it more of a classification algorithm than a regression one. Specifically, it is used when the dependent variable is binary, having two potential outcomes. LR models the likelihood that each input belongs to a specific category, outputting a value between 0 and 1. The probability $p$ is defined by:

$$\ln \frac{p}{1-p} = \beta_0 + \beta_1 x_1 + \beta_2 x_2 + \cdots + \beta_n x_n \tag{3}$$

where $\beta_i$ is the weight coefficient for the explanatory variables.

LR serves as a primary tool for establishing a decision boundary in binary classification, enabling the prediction of the corresponding class for a new set of features. An intriguing aspect of logistic regression lies in its employment of the sigmoid function as the estimator for the target class.

**KN.**   KN is a nonparametric supervised learning algorithm that predicts the class or value of unknown data points based on the $K$ most similar data points in the training dataset. The

similarity between the data points is typically measured using Euclidean distance. The performance of KN depends on the right choice of a value of *K*.

**NB.**   NB stands out as a well-known supervised ML technique leveraging Bayes' theorem and assuming the independence of input features. It does not represent a singular algorithm but rather a family of algorithms united by the common principle that the classification of any two features is independent.

**TX.**   TX consists of an encoder and decoder that are created to process sequential input data, such as natural language for text translation and summarization. As a DL model, TX employs an attention mechanism that differentially weights the importance of each word in the input text [37]. The encoder layer consists of a multi-head attention network and a feed-forward network. In this study, we used the encoder of TX and set the number of attention heads and layers to 4 and 4, respectively [36].

**CNN.**   The CNN consists of convolutional and pooling layers. In the convolutional layer, significant features can be extracted based on filters. The pooling layer provides robust prediction with respect to pattern modification and suppresses overfitting by compressing the information. As proposed in a previous study [38], we applied CNN architectures consisting of two convolutional layers and two max-pooling layers.

**bLSTM.**   Recurrent neural networks are useful for making predictions about interdependent data such as time-series, but these networks are not suitable for learning long-term dependencies because of gradient disappearance and explosion. To address this issue, LSTM introduces gate structures and memory cells to expand the LSTM units in two directions, called BiLSTM approach [39].

The seven ML and three DL methods were implemented using scikit-learn [40] and PyTorch [23], respectively. For DLs we used Adam optimizer with binary cross-entropy function. The hyperparameters for each method were optimized during the training process, as shown in S1 Table. We utilized grid search coupled with 5-fold CV to meticulously refine these hyperparameters, and a comprehensive overview of this process can be found in our previous study [36].

## Meta-classifier

By connecting seven ML classifiers and three DL ones to 22 encoding methods, we generated 168 single-feature models via 5-fold CV on the training dataset. Then, we constructed the meta-classifier that stacks the predicted probabilities resulting from each single-feature model to attain accurate prediction [41, 42]. The meta-classifier is trained via 5-fold CV so that the final predicted probabilities could fit the class labels. We investigated the seven meta-classifiers including LR, SVM, LGBM, XGB, RF, NB, and KN.

It is critically important to select single-feature models out of many 168 models. Generally, the feature selection is a dimensionality reduction technique that selects a subset of features that present the best predictive power. In this study, note that the features correspond to single-feature models. Feature selection is used to prevent overfitting and to improve interpretability with fewer features. There are multiple methods to select features: Iterative changes of features subset sequentially adds or removes the features until no improvement occurs in prediction; Ranking of features based on intrinsic characteristic such as mutual information is used to select the top few ranked features; ML-learned feature importance during the training process is used to rank features. We propose the following three feature selection methods that rank the single-feature models in terms of prediction performance (e.g., AUC) or their importance (weight coefficient) during the training process.

### Sequential addition of single-feature models based on the AUC ranking (SAAUC)

The SAAUC selects the top X single-feature models in the descending order of the AUC values during the training process, where X is incremented by one. LR, SVM, LGBM, XGB, RF, NB, and KN are used as a meta-classifier.

### Sequential addition of single-feature models based on the AWCLR ranking (SAWC)

The SAWC selects the top X single-feature models in the descending order of the AWCLR that corresponds to the importance of each single-feature model. LR is employed as a meta-classifier, because the importance is explicitly defined as the weight coefficients of LR by Eq (3), which indicate the contribution of each single-feature model to accurate prediction.

### Sequential deletion of single-feature models based on the iterative AWCLR ranking (SDIWC)

The SDIWC selects the top X models in the descending order of the AWCLR, while iteratively updating the AWCLR ranking as follow, because the AWCLR ranking can vary with the employed single-feature model subset: (1) Calculate the AWCLR of all single-feature models by a meta-classifier of LR. (2) Sort the single-feature models in the descending order of the AWCLR. (3) Remove the single-feature model with the lowest AWCLR. (4) Input the remaining single-feature models into the LR meta-classifier. (5) Repeat (2–4) until the number of the remaining single-feature models becomes one.

### Evaluation

The effectiveness of the proposed predictive models was assessed using seven established statistical measures, each offering different insights into the prediction performance. It includes accuracy (ACC), sensitivity (SEN), specificity (SPE), precision (PRE), Matthews correlation coefficient (MCC), area under the receiver operating characteristic curve (AUC) and area under the precision-recall curve (AUPRC). Comprehensive descriptions and mathematical formulations of these measures are available in previous studies [36]. A threshold value that determines whether the probability scores are classified into positive and negative samples is adjusted so as to maximize MCC.

## Results and discussion

### Single-feature models: Construction and independent evaluation

Generally, the performance of ML and DL classifiers depends on their learning algorithms and encoding methods. The performance of these classifiers, trained with the same encoding method, varies with their algorithms. To consider a variety of features produced by a combination of different learning methods with encodings, we constructed 168 single-feature models by combining seven ML and three DL classifiers with 22 distinct feature encodings including language models. For instance, the LR model trained with the AAC encoding is regarded as a single-feature model. All the single-feature models were trained on the training dataset via 5-fold CV, then evaluated using an independent test dataset.

As depicted in Fig 2, we characterized the prediction performance (AUC, MCC) of 168 single-feature models on the training and test datasets. Details in their prediction performances are shown in S2 Table. The performance trend of each single-feature model was almost

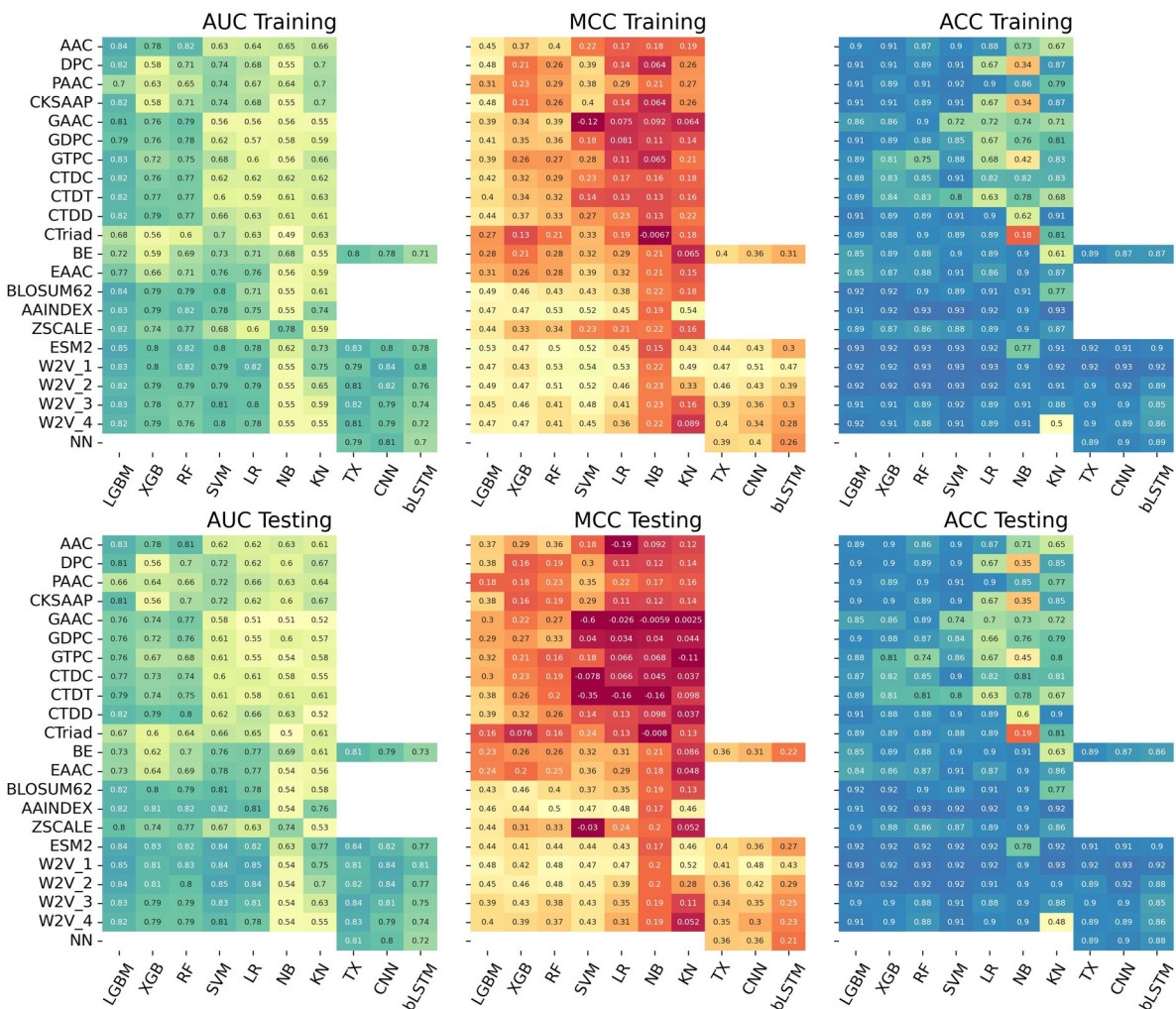

**Fig 2. Heatmap of the prediction performance of the 168 single-feature models.** The models were built by employing 7 ML classifiers and 3 DL classifiers with different 21 encoding methods. Mathew's correlation coefficient (MCC), accuracy (AAC) and area under the curve (AUC) on the training and test datasets are illustrated for each single-feature model. "-1" indicates that no learning model is built.

consistent between the training and test datasets, which indicates that the models are well trained. The LGBM showed high performance (AUC > 0.8) with respect to composition-based encodings (AAC, DPC, CKSAAP, GAAC, GDPC, GTPC, CTDC, CTDD), position-based encodings (BLOSSUM62, AAINDX, ZSCALE) and the language models (ESM-2, W2V). Particularly, the LGBM with ESM-2 provided remarkable performance: AUCs of 0.847 and 843 and MCCs of 0.528 and 0.438 in the training and test datasets, respectively. The RF indicated high performance with AAINDEX, ESM-2 and W2V. The boundary-based models (SVM, LR) presented high performance with BLOSSUM62 and language models (ESM-2, W2V). On the other hand, the NB and KN provided less performance with respect to all the encodings.

In the DL methods, TX and CNN with the language models (ESM-2, W2V) presented a high performance; bLSTM showed a lower performance than TX and CNN. Use of the language models could enhance the performance more than use of BE and NN. The TX models with W2V-3 and W2V-4 and the CNN models with W2V-1 and W2V-2 presented high

performance. The performances of TX and CNN with the language models were competitive to those of LGBM.

To clearly illustrate the prediction capability of all the 168 single-feature models, we ranked them according to their AUC values on the training dataset (S3 Table). The LGBM with ESM-2 was the first ranked model; most of the LGBM-based models were placed at high ranks. The second ranked model was the CNN with W2V-1. In general, the language models (ESM-2, W2V) and composition-based encodings were placed at high ranks for many ML and DL methods.

## Feature selection for meta-classifier construction

Since stacking of single-feature models is typically expected to enhance the prediction performance [41, 42], we proposed the three feature selection methods: SAAUC, SAWC, and SDIWC (see Methods). First, by using the SAAUC method, we stacked the top X single-feature models-generated probability vectors according to the AUC ranking, then inputted them into a meta-classifier, including LR (Fig 3), SVM, LGBM, XGB, RF, KN, and NB (S1 Fig). The meta-classifier was trained with the training dataset, while incrementing X by one. If a change in AUC with respect to X has one peak or a saturation curve, it would be easy to select the optimal X, which corresponds to the peak or just attains saturation. For LR, the AUC and MCC values tended to increase with an increase in X, while it went up and down with respect to X. The AUC of the top 2 model dropped as CNN-W2V_1 was added to the top 1 of LGBM-ESM-2, suggesting that the CNN-W2V_1 and LGBM-ESM-2 are incompatible pairs. However, the prediction performance could be improved by an increase in the number of single-feature models. The other meta-classifiers also showed the same trend in the AUC and MCC with respect to X. Since one single-feature model produces one score for each sequence, the dimension of the produced score vectors (X) is much fewer than the sample number. It may cause overfitting, but the prediction performances of all the seven classifiers in test were rather consistent with that in the validation and overfitting was not observed, showing the robustness and generality of the prediction model. Out of the seven meta-classifiers, LR was the best predictor in terms of prediction performance on both the training and test datasets. Thus, we determined LR as the meta-classifier (S4 Table).

## AWCLR-based feature selection

To suppress the fluctuations in AUC and MCC and reveal the effectiveness of stacking, we focused on the importance of each single-feature model during training, i.e., we applied the SAWC method to a series of the LR-based meta-classifiers (X = {1, 2,. . ., 168}) and calculated the weight coefficients defined by Eq (3) as the importance. Specifically, we retrieved the absolute weight coefficients of LR (AWCLR) of 168 single-feature model from the meta-classifier with X = 168, as shown in Fig 4. Interestingly, the AWCLR-based ranking of the single-feature models differed from the AUC-based ranking of them. We sorted the single-feature models in the descending order of AWCLR (S5 Table) and stacked the top X single-feature models according to the AWCLR ranking to evaluate their performance on the training and test datasets, as shown in Fig 5. The AUC initially dropped, then it increased, achieving a plateau at the top 58 (X = 58). The AWCLR ranking-based method (SAWC) was found to be an indicator superior to the SAAUC for feature selection. The fluctuation in performance were suppressed at X of < 80, but it still fluctuated at X of >80.

To further improve the fluctuation in the performance, we applied the SDIWC method to the series of the LR-based meta-classifiers. The prediction performance of the top X models varied smoothly as shown in Fig 6. The AUC initially dropped, then it increased, achieving a

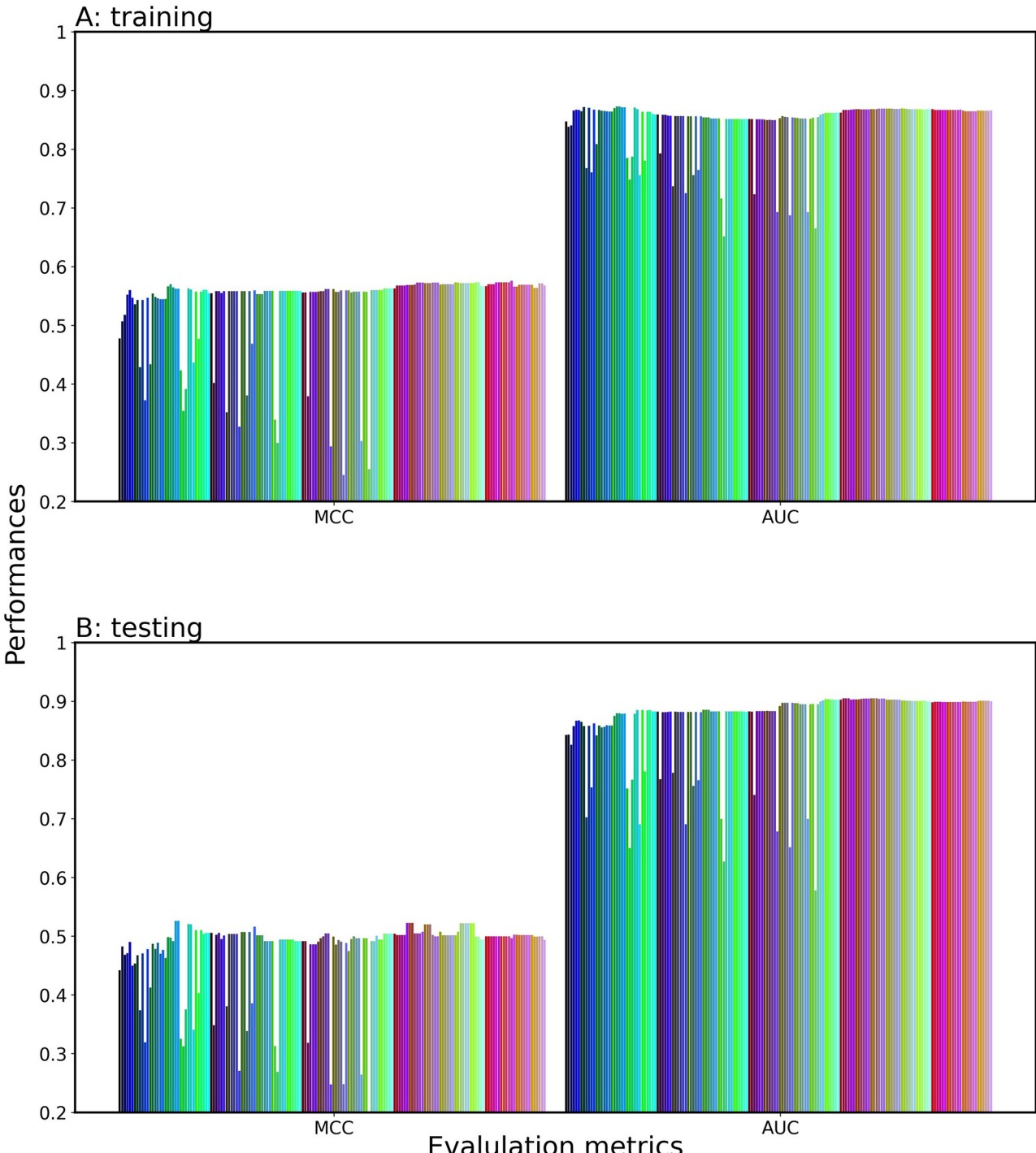

**Fig 3. Prediction performance of the SAAUC-stacked classifier consisting of the top X single-feature models.** LR is employed as the meta-classifier and X is incremented by one from one to 168. (A) validation dataset; (B) test datasets.

plateau at X = 58. The initial fall would be caused by a compatibility of the top 2 models (LGBM-ESM-2 and LGBM-W2V_4). This AUC curve enabled us to readily select the optimal single-feature models. Table 1 summarized the performance of the top X model with the highest AUC for the three feature selection methods. We selected the top 58 model as the best

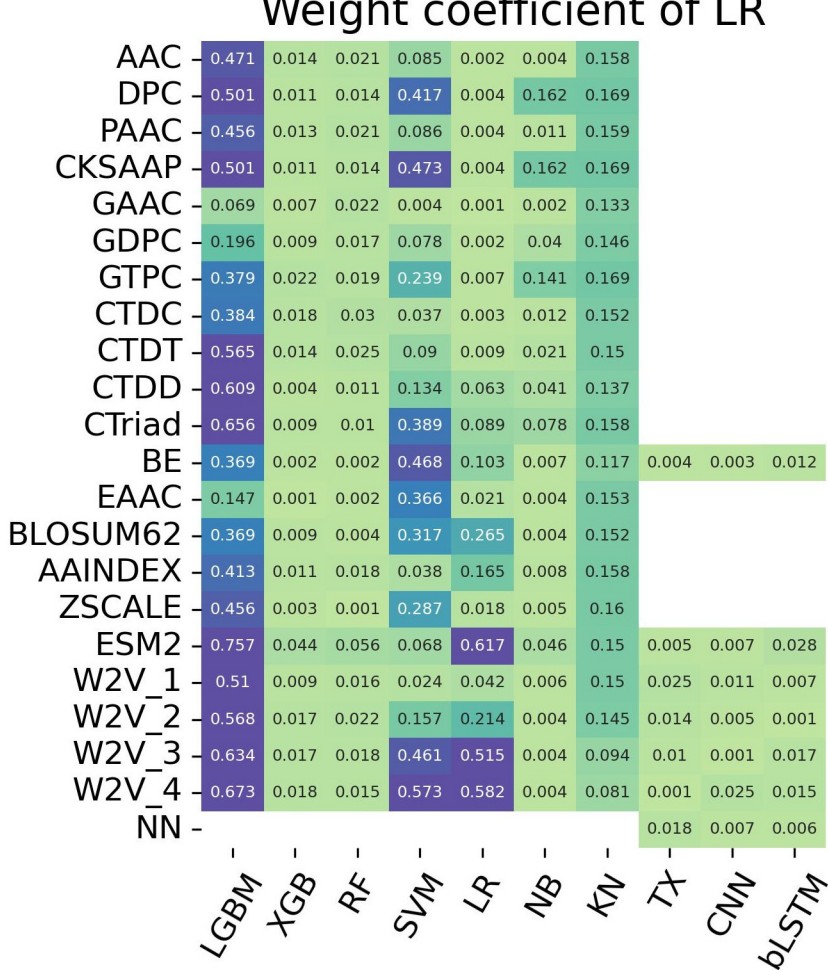

**Fig 4. Heatmap of the AWCLR of the 168 single-feature models.** The models were built by employing 7 ML classifiers and 3 DL classifiers with different 21 encoding methods. The AWCLR indicates the contribution of each single-feature model to the prediction performance.

model, and designated it PredIL13 (top 58). PredIL13 achieved a consistent level of performance between the training and test, indicating the generality and robustness on the test dataset. The resultant two meta-classifiers built by SAWC and by SDIWC had the same optimal number (58) of single-feature models. This number might be obtained by chance, because the AWCLR ranking changed with a change in a subset of the employed single-feature models.

When compared to the first ranked single-feature model (LGBM-ESM-2), the stacking method greatly enhanced the prediction performance, demonstrating the effectiveness of the SDIWC method. In the independent dataset, LGBM-ESM-2 showed SEN of 0.286, SPE of 0.989, PRE of 0.796, ACC of 0.920, MCC of 0.438, AUC of 0.843, and AUPRC of 0.573; PredIL13 achieved SEN of 0.327, SPE of 0.993, PRE of 0.849, ACC of 0.928, MCC of 0.499, AUC of 0.899, and AUPRC of 0.635.

## Importance of each single-feature model

To understand the contribution or importance of each single-feature model to prediction performance, we analyzed the heatmap of AWCLR (Fig 4) in comparison with the heatmap of

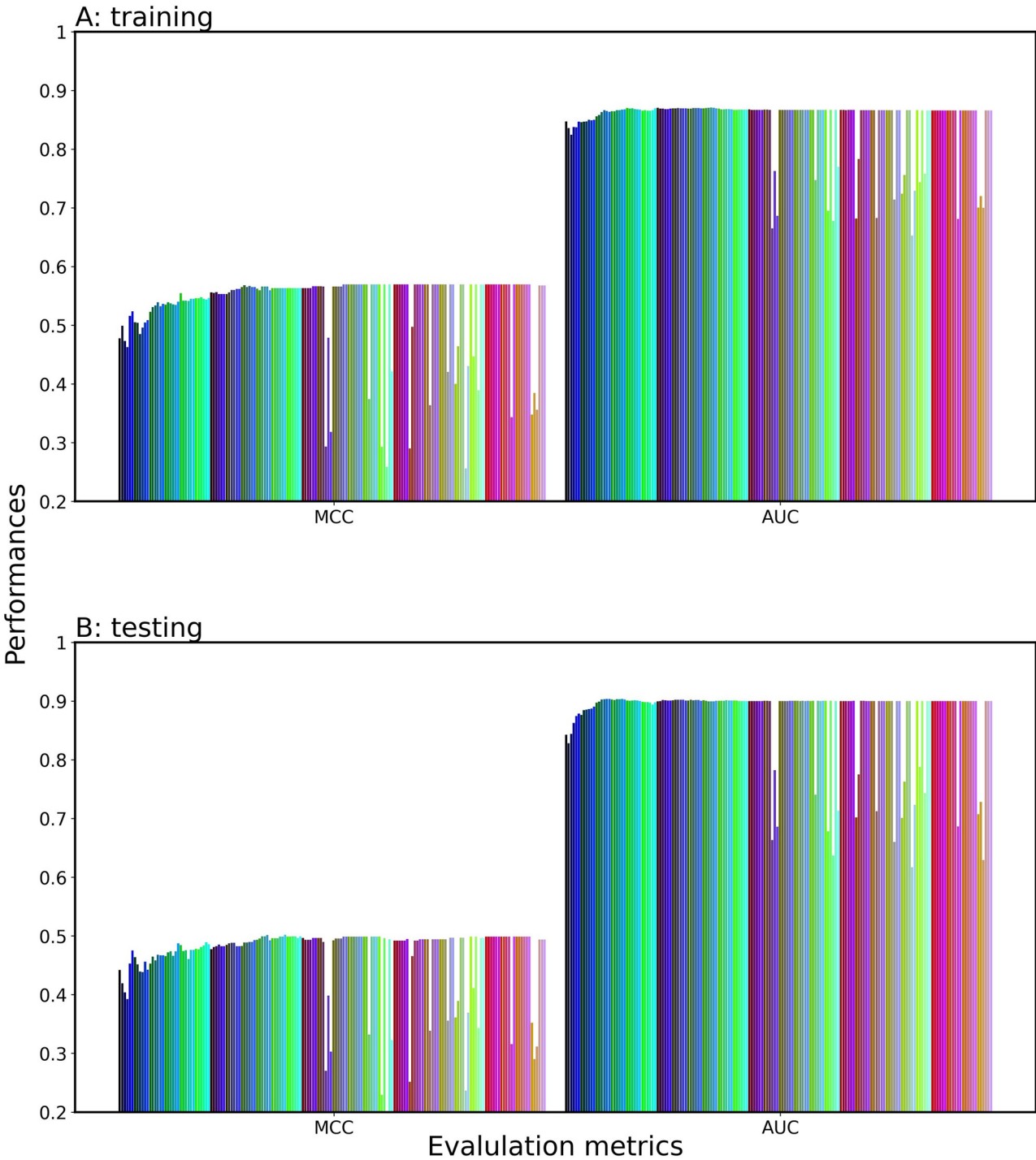

**Fig 5. Prediction performance of the SAWC-stacked classifier consisting of top X single-feature models.** LR is employed as the meta-classifier and X is incremented by one from one to 168. (A) validation dataset; (B) test datasets.

AUC (Fig 2). Interestingly, all the DL-based models dropped to lower ranks in the AWCLR ranking, which was exemplified by the fact that the CNN-W2V_1 dropped from the second-best model in the AUC ranking to the 116th model in the AWCLR ranking and the highest ranked DL models (bLSTM-ESM-2, TX-W2V_1, CNN-W2V_4) in the AUC ranking fell to

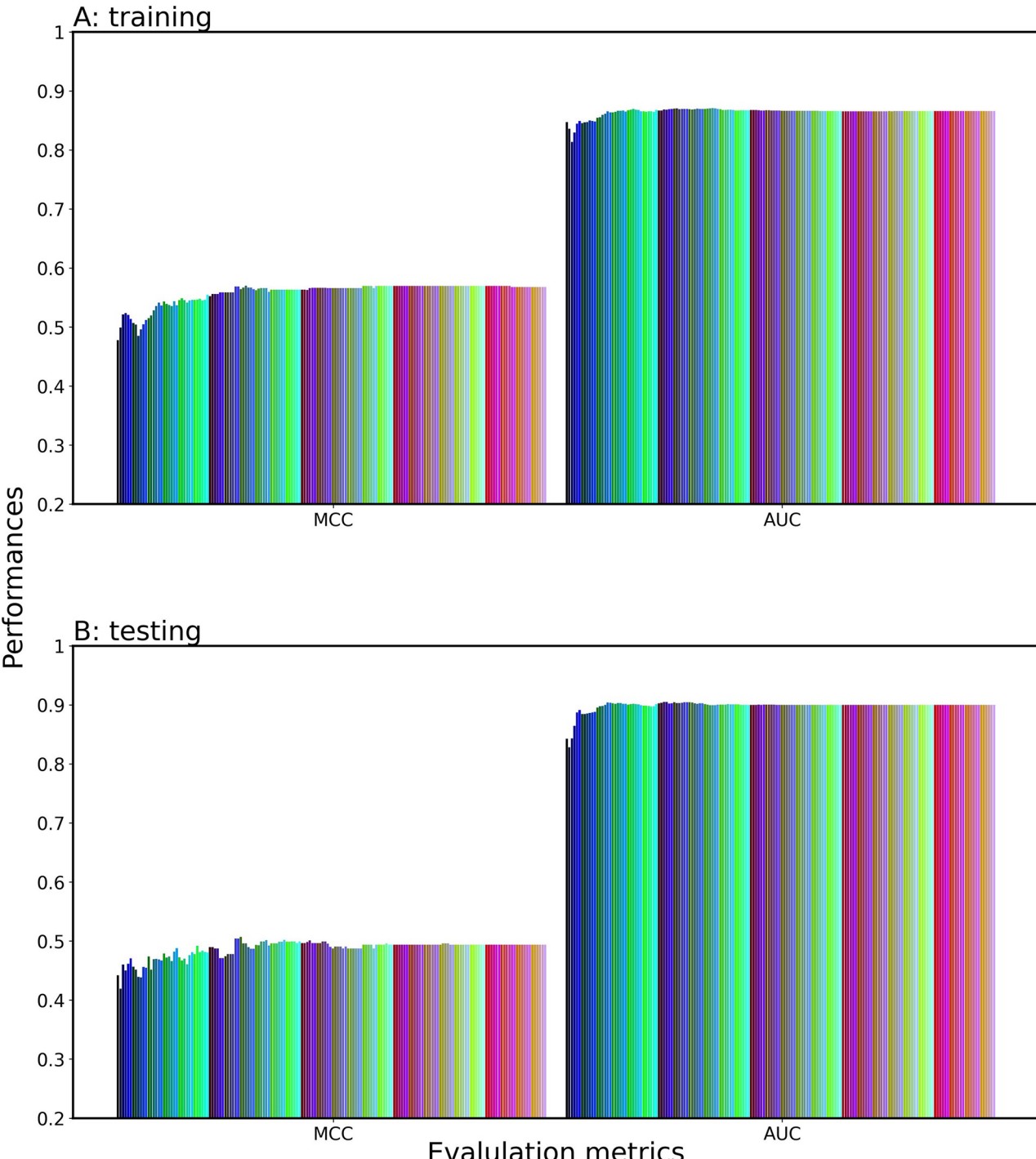

**Fig 6. Prediction performance of the SDIWC-stacked classifier consisting of top X single-feature models.** LR is employed as the meta-classifier and X is incremented by one from one to 168. (A) validation dataset; (B) test datasets.

the 82th, 83th, and 84th in the AWCLR ranking. Consequently, the top 58 model did not include any DL methods. ML and DL could not work together for enhanced performance, suggesting some incompatibilities between the probability score vectors generated by ML and DL models. Many LGBM-based single feature models and SVM- and LR-based models with the

**Table 1. The best LR-based meta-classifier that provides the highest AUC for three feature selection methods.** The validation dataset is used for training.

| Feature selection | Top X | | SEN | SPE | PRE | ACC | MCC | AUC | PRAUC |
|---|---|---|---|---|---|---|---|---|---|
| SAAUC | 21 | Training | 0.408 | 0.994 | 0.871 | 0.937 | 0.57 | 0.873 | 0.628 |
| | | Test | 0.305 | 0.996 | 0.899 | 0.928 | 0.497 | 0.88 | 0.628 |
| SAWC | 58 | Training | 0.436 | 0.988 | 0.818 | 0.935 | 0.566 | 0.871 | 0.619 |
| | | Test | 0.327 | 0.993 | 0.849 | 0.928 | 0.499 | 0.899 | 0.635 |
| SDIWC | 58 | Training | 0.436 | 0.988 | 0.818 | 0.935 | 0.566 | 0.871 | 0.619 |
| | | Test | 0.327 | 0.993 | 0.849 | 0.928 | 0.499 | 0.899 | 0.635 |
| SDIWC | 16 | Training | 0.388 | 0.993 | 0.850 | 0.934 | 0.547 | 0.865 | 0.615 |
| | | Test | 0.302 | 0.996 | 0.894 | 0.928 | 0.491 | 0.906 | 0.646 |

language models (ESM-2, W2V) were placed at high ranks in the AWCLR ranking, indicating that the tree-based and boundary-based models are effective in the stacking approach.

Feature selection methods other than the AWCLR, which represents the contribution of each single-feature model (feature) to the log-odds of binary classification, are known. For example, SHapley Additive exPlanations (SHAP) analysis provides a more nuanced view by showing how each feature contributes to individual predictions (S2 Fig). In this study we used the AWCLR as it has a theoretical intelligible basis. SHAP analysis will be considered elsewhere.

### Analysis of the top 58 single-feature models

Since the number of the single-feature models included in the top 58 was relatively large, indicating model complexity, it would be better if we reduce the number of the selected models. To further conduct the feature selection, we considered the probability distributions generated by the top 58 single-feature models for the positive and negative samples during training, as shown in Fig 7. The top 29 models were mainly based on LGBM and SVM, and the probability scores of them were clearly separated between the positive and negative samples. On the other hand, the probability scores from the 30th to 58th models were not clearly separated. Especially, in the KN-based single-feature models (KN-CKSAAP, KN-DPC, KN-ZSCALE, KN-PAAC, KN-Ctriad, KN-AAC, KN-EAAC, KN-CTDC, KN-BLOSSUM62, KN-GDPC, KN-W2V_2, KN-CTDT), their probability profiles were much overlapped between the positive and negative samples. In the NB-based models (NB-CKSAAP, NB-DPC, NB-GTPC), the probability scores for negative samples were broadly scattered from 0 to 1. These KN- and NB-based models were confirmed to present low values of AUC, MCC, and ACC (Fig 2), indicating that their contributions are rather small.

To further select the single-feature models, we proposed deleting the single-feature models with ACC < 0.902 based on the following mechanism. If all the peptides are predicted to be negative, we obtain an ACC value of more than 0.902 (= 2908/(2908+313)) in our imbalanced dataset (See Dataset preparation). Following this deletion process, we selected 72 single-feature models with ACC ≥ 0.902 out of 168 and applied SDIWC to these 72 models, as shown in Fig 8. The AUC increased after X = 3, and achieved a peak at X = 16 during training, which intelligibly selected the top 16 model as the best model. It was named as PredIL13 (top 16). Looking at the top 16 single-feature models (Table 2), composition-based features (AAC, PAAC, DPC, CKSAAP, CTDD) were found to be effective in detecting the IL13-inducing activity. Particularly, the ESM-2 and multi-Kmer (1, 2, 3, and 4) W2V were very effective, which enabled the prediction model to perceive a wide range of contextual information or amino acid sequence patterns at different scales. The feature selection was also effective in enhancing the interpretability of the meta-classifier.

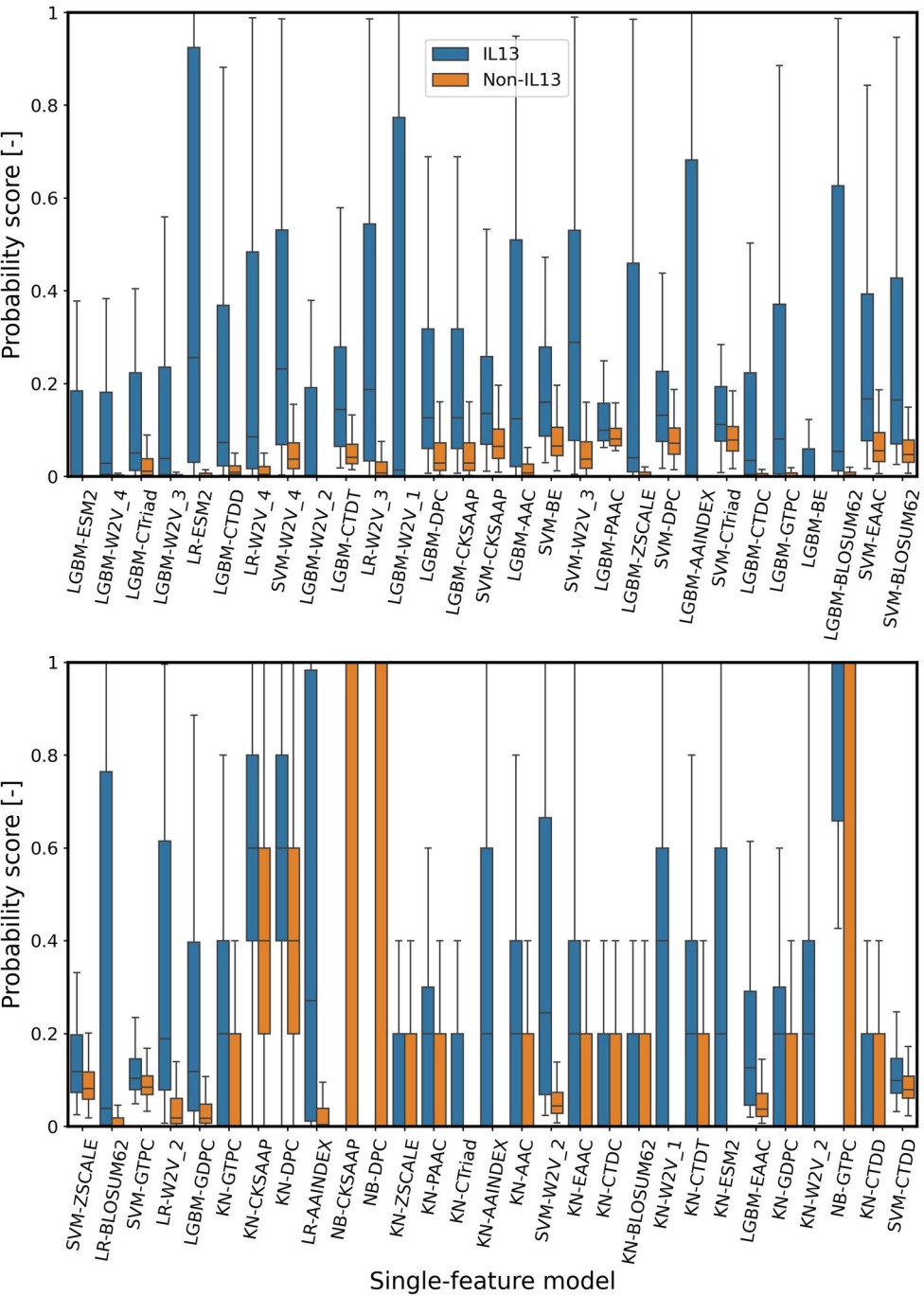

**Fig 7. Boxplots of the probability distributions of the top 58 single-feature models for on the training dataset.** The single-feature models were arranged in the descending order of AWCLR from the left to the right. (Upper panel) The models from the top 1 to top 29. (Lower panel) the models from the top 30 to 58.

## Comparison of PredIL13 with existing predictors

To characterize the proposed methods from a fair perspective, we benchmarked their performance against two state-of-the-art predictors, IL13Pred and iIL13Pred, using the same test dataset. The single-feature model of LGBM-ESM-2 showed a little higher AUC than IL13Pred

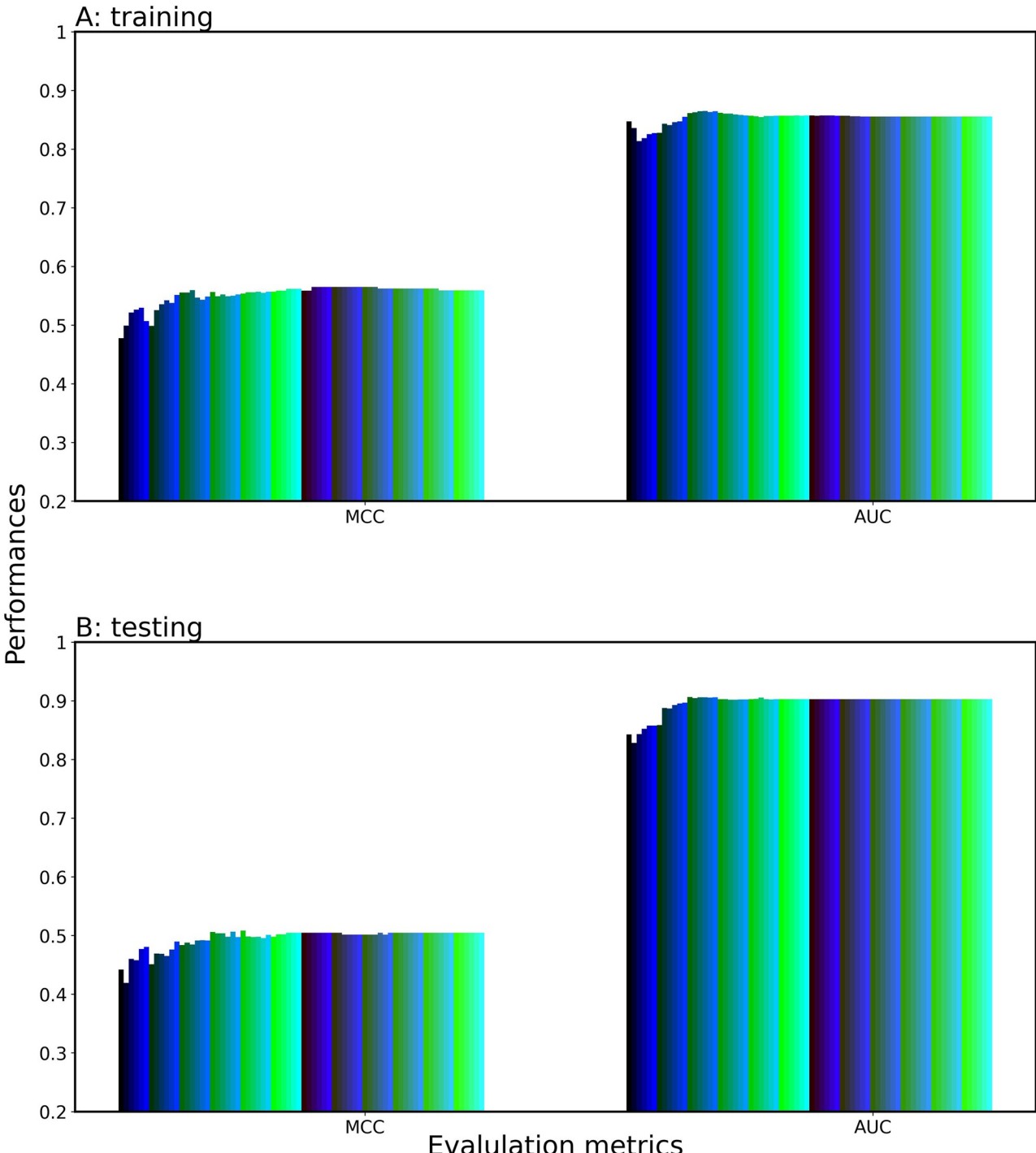

**Fig 8. Prediction performance of the SDIWC-stacked classifier consisting of top X single-feature models.** The 72 single feature models with ACC > 0.902 on the training dataset were employed. LR is employed as the meta-classifier and X is incremented by one from one to 72. (A) validation dataset; (B) test datasets.

**Table 2. Top 16 single-feature models.**

| | | | |
|---|---|---|---|
| LGBM-ESM-2 | LGBM-AAC | LGBM-W2V_1 | LGBM-W2V_3 |
| LGBM-AAINDEX | LGBM-W2V_4 | LGBM-CKSAAP | LGBM-DPC |
| LGBM-W2V_2 | LGBM-CTDD | SVM-W2V_3 | SVM-W2V_4 |
| LR-ESM-2 | SVM-DPC | SVM-CKSAAP | LGBM-PAAC |

**Table 3. Comparison of our proposed PredIL13 with the-state-of-the-art method.**

| Method | SEN | SPE | PRE | ACC | MCC | AUC | PRAUC |
|---|---|---|---|---|---|---|---|
| IL13Pred | 0.718 | 0.7302 | - | 0.7287 | 0.3 | 0.8 | - |
| iIL13Pred | **0.746** | 0.7577 | - | 0.7566 | 0.33 | 0.83 | - |
| PredIL13 (top 58) | 0.327 | 0.993 | 0.849 | 0.928 | 0.499 | 0.899 | 0.635 |
| PredIL13 (top 16) | 0.302 | **0.996** | **0.894** | 0.928 | 0.491 | **0.906** | **0.646** |
| PredIL13 (top 14) | 0.337 | 0.994 | 0.862 | **0.930** | **0.511** | 0.897 | 0.625 |
| LGBM-ESM-2 | 0.286 | 0.989 | 0.796 | 0.920 | 0.438 | 0.843 | 0.573 |

Note: '-' indicates that data is not available. The values in bold indicate the top value.

and iIL13Pred, despite its relative simplicity. This observation emphasized that the latest language model ESM-2 was effective in extracting critically important patters and distributions of amino acids.

Our proposed meta-classifier, PredIL13, yielded impressive results (Table 3). While PredIL13 (top 58, top 16) showed lower sensitivity than IL13Pred and iIL13Pred, PredIL13 distinctly surpassed IL13Pred and iIL13Pred in key performance indicators of SPE, ACC, MCC, and AUC. Especially, PredIL13 (top 16) took a remarkable advantage: 9.2% higher AUC than iIL13Pred. PredIL13 made a great progress in distinguishing IL13-inducing peptides and demonstrated its ability to capture preferred patterns and distributions from the sequence data. It improved state-of-the-art predictors in the field of IL13-inducing peptide classification.

Since ESM-2 required a large amount of memory (60GB in our study), we investigated if the ESM-2 encoding is indispensable. We removed the two single-feature models that possessed EMS2, i.e., LGBM-ESM-2 and LR-ESM-2, out of the top 16 models to build PredIL13 (top14). While PredIL13 (top14) slightly decreased AUC and AUPRC on the test dataset, it increased ACC and MCC. The ESM-2-free meta-classifier showed competitive performance to PredIL13(top16), indicating that the ESM-2-free meta-classifier was sufficiently effective despite the top 1 model (LGBM-ESM-2) being omitted. We recommend the users to employ the ESM-2-free model (top 14), when their servers are running out of memory.

Note that SEN decreased in PredIL13 compared to IL13Pred and iIL13Pred. We considered that a low value of SEN resulted from the evaluation process of a meta-classifier. During the evaluation, the number of the positive samples predicted to be a negative class (FP) decreased, while that of the negative samples predicted to be a positive class (FN) increased. It is because the employed datasets are imbalanced and the threshold value, which classifies the probability scores into the positive and negative ones, is set to a low value when the evaluation process maximizes MCC on the training dataset.

## Application to SARS-CoV-2 spike proteins

To demonstrate the prediction capability of PredIL13, we tested it with the experimentally validated IL-13-inducing peptides that derived from the immune epitope database [12] [13]. IL13Pred and iIL13pred accurately predicted 19 and 37 peptides as IL13-inducing peptides out of a total 68 experimentally validated IL-13-inducing peptides, indicating SENs of 0.28 and 0.54, respectively. On the other hand, our PredIL13 correctly predicted 29 peptides as IL13-inducers (S6 Table), presenting a SEN of 0.43. PredIL13 predicted IL13-inducing peptides more accurately than IL13Pred, but less than iIL13Pred. Such a decreased SEN would be caused by a high value of SPE and PRE (Table 3). PredIL13 takes an advantage in increasing SPE and PRE.

## Limitation

One limitation of this study is that the number of experimentally validated IL-13-inducing peptides is small. Thus, we need to construct a larger-scale dataset to increase the prediction performance. Furthermore, using the large dataset we construct a generative AI to design de novo peptide sequences and to ensure the generated sequences are biologically functional and potentially beneficial for medication.

## Conclusion

We have proposed PredIL13, a novel, efficient computational methodology based on the stacking or ensemble strategy, specifically designed for predicting human IL13-inducing peptides. This method stacked 168 single-feature ML/DL models by combining their probability scores, trained the LR-based meta-classifier with the combined probability vectors, and selected a subset of the single-feature models that maximized the prediction performance. We proposed the SDIWC method to efficiently select the optimal single-feature models. From each trained single-feature model, we retrieved the AWCLR, the importance of each single-feature model, and sorted the single-feature models in the descendent order of AWCLR. The SDIWC method selected the top 58 models (IL13Pred (top 58)), while iteratively updating the AWCLR ranking. To further select the features, we applied the SDIWC method to the single-feature models with ACC $\geq$ 0.902. Finally, we obtained the LR-based meta-classifier that consisted of the top 16 single-feature models (IL13Pred (top 16)). The SDIWC method enabled us to intelligibly select the optimal single-feature models.

Importantly, the proposed single-feature model selection method revealed crtical fetaures responsible for IL13-inducing activity. Looking at the top 16 single-feature models, the linguistic approaches of ESM-2 and W2V, employed by the LGBM and SVM models, took a great advantage in improving the model's performance. The ESM-2 and multi-Kmer W2V enabled the model to perceive a wide range of contextual information or amino acid sequence patterns at different scales. Particularly, ESM-2 presented the most important feature for identifying the IL13-inducing activity. Interestingly, in the meta-classifier, the high ranked DL methods in the AUC ranking moved to lower ranks in the AWCLR ranking, indicating the importance of DL methods decreased, despite the high performance of the DL-based single feature models. The resultant PredIL13 did not include any DL methods.

The PredIL13 outperformed the-state-of-the-art predictors, thus is an invaluable tool in accelerating the detection of IL13-inducing peptide in human. PredIL13 represents a computational strategy for the high-throughput and accurate prediction of human IL13-inducing peptides. The proposed selection method can be applied to a variety of sequence-based functional prediction tasks, such as peptide therapeutics.

## Supporting information

**S1 Fig. Prediction performance of six meta-classifiers built by the SAAUC method on the validation and test datasets.** (A) LGBM; (B) XGB; (C) RF; (D) SVM; (E) NB (F) KNN.
(PDF)

**S2 Fig. SHAP analysis of the SDIWC-stacked classifier consisting of top 72 single-feature models.**
(PDF)

**S1 Table. Hyperparamter tuning for ML and DL.**
(XLSX)

**S2 Table. Prediction performance of 168 single feature models on the training and test datasets.**
(XLSX)

**S3 Table. Ranking of 168 single-feature models in the descending order of AUC on the training dataset.**
(XLSX)

**S4 Table. Prediction performance of seven meta-classifiers on the training and test datasets.** The SAAUC method is employed. The validation dataset is used for training.
(XLSX)

**S5 Table. Ranking of 168 single-feature models in the descending order of AWCLR on the training dataset.**
(XLSX)

**S6 Table. Prediction of experimentally validated IL13-inducing peptides by PreIL13 (top16).**
(XLSX)

## Author Contributions

**Conceptualization:** Hiroyuki Kurata.

**Data curation:** Md. Harun-Or-Roshid.

**Formal analysis:** Hiroyuki Kurata, Md. Harun-Or-Roshid.

**Funding acquisition:** Hiroyuki Kurata.

**Investigation:** Md. Harun-Or-Roshid.

**Methodology:** Hiroyuki Kurata, Md. Harun-Or-Roshid.

**Project administration:** Hiroyuki Kurata.

**Resources:** Md. Harun-Or-Roshid, Sho Tsukiyama, Kazuhiro Maeda.

**Software:** Hiroyuki Kurata, Sho Tsukiyama, Kazuhiro Maeda.

**Supervision:** Hiroyuki Kurata.

**Validation:** Sho Tsukiyama, Kazuhiro Maeda.

**Writing – original draft:** Hiroyuki Kurata.

**Writing – review & editing:** Md. Harun-Or-Roshid, Kazuhiro Maeda.

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
