## [Decision Letter · Decision Letter 0]

2 Jul 2024

PONE-D-24-20864PredIL13: stacking a variety of machine and deep learning methods by weight coefficient-based single-feature model selection for identifying IL13-inducing peptidesPLOS ONE

Dear Dr. Kurata,

Thank you for submitting your manuscript to PLOS ONE. After careful consideration, we feel that it has merit but does not fully meet PLOS ONE’s publication criteria as it currently stands. Therefore, we invite you to submit a revised version of the manuscript that addresses the points raised during the review process.

**Major Revision**

We look forward to receiving your revised manuscript.

Kind regards,

Shahid Akbar, PhD

Academic Editor

PLOS ONE

Journal Requirements:

https://doi.org/10.1186/s12859-023-05248-6

In your revision ensure you cite all your sources (including your own works), and quote or rephrase any duplicated text outside the methods section. Further consideration is dependent on these concerns being addressed.

This work is supported by Japan Society for the Promotion of Science(JSPS) with grant number 22H03688. In relation to this, the funder had no role in study design, data collection and analysis, decision to publish, or preparation of the manuscript.

5. Thank you for uploading your study's underlying data set. Unfortunately, the repository you have noted in your Data Availability statement does not qualify as an acceptable data repository according to PLOS's standards.

Reviewers' comments:

Reviewer's Responses to Questions

**Comments to the Author**

1. Is the manuscript technically sound, and do the data support the conclusions?

Reviewer #1: Partly

Reviewer #2: Partly

2. Has the statistical analysis been performed appropriately and rigorously? 

Reviewer #1: Yes

Reviewer #2: Yes

3. Have the authors made all data underlying the findings in their manuscript fully available?

Reviewer #1: Yes

Reviewer #2: Yes

4. Is the manuscript presented in an intelligible fashion and written in standard English?

Reviewer #1: Yes

Reviewer #2: Yes

5. Review Comments to the Author

**Reviewer #1:** In the abstract, the authors should mention the feature encoding methods , more specifically the protein language model.

2. In The introduction, the literature section is very poor, the authors should add some computational methods based papers related to PredIL13.

3. The contribution should be clearly mentioned in points at the end of introduction section.

4. The authors should mention the provided the hyper-parameters used for training the used machine learning and deep learning in the form of a table.

5. The code of the interpretation methods used in paper should be provided for the reimplementation purposes.

6. For the reader concerns the authors are advised to add the recent predictors such as AIPs-SnTCN, DeepAVP-TPPred, Deepstacked-AVPs,iAFPs-Mv-BiTCN and pAVP_PSSMDWT-EnC models

7.The feature vector size of the encoding methods should be provided in the paper.

8. What are limitations of the proposed model.

**Reviewer #2:** 1. How the authors handle the imbalance data while training the model. Did they used any oversampling or under sampling approach, authors should clearly mention.

2. What is justification of using the machine learning models in the proposed model? As there are several other training models available in the literature.

3. As the authors used several feature extraction methods to formulate peptide sequence. However, It should be convincing if the authors highlight the high contributory features using SHAP analysis.

4. What should be the future direction of the proposed model.

6. PLOS authors have the option to publish the peer review history of their article (what does this mean?). If published, this will include your full peer review and any attached files.

Reviewer #1: No

Reviewer #2: No

---

## [Author Response · Author response to Decision Letter 0]

15 Jul 2024

Reviewer #1:

Q1.In the abstract, the authors should mention the feature encoding methods, more specifically the protein language model. 

A1. Thank you for your useful comment. We mentioned the ESM-2 protein language model in Abstract. In addition, we revised the title to attract many readers as follows: PredIL13: stacking a variety of machine and deep learning methods with ESM-2 language model for identifying IL13-inducing peptides.

Q2. In The introduction, the literature section is very poor, the authors should add some computational methods based papers related to PredIL13.

A2. A thorough search of the literature reveals that IL-13Pred (2022) is the first to predict the IL-13 inducing and non-inducing peptides from its amino acid sequence. To data only a few predictors have been presented. We mentioned that the development of IL-13 predictors has just begun in Introduction. 

Q3. The contribution should be clearly mentioned in points at the end of introduction section.

A3. According to the suggestions, we mentioned the following points at the end of Introduction: (1) PredIL13 is a novel meta-classifier that effectively stacks a variety of single-feature models by using logistic regression to predict IL13-inducing peptides; (2) The SDIWC method is proposed to effectively select the 16 optimal single-feature models out of the 168 single-feature models according to their importance; (3) Language models including ESM-2 are effective in increasing the prediction performance and greatly outperforms state-of-the-art methods in terms of prediction performances.

Q4. The authors should mention the provided the hyper-parameters used for training the used machine learning and deep learning in the form of a table.

A4. We provided the hyper-parameters used for training the used machine learning and deep learning in Table S1 at the end of the ML and DL classifiers section.

Q5. The code of the interpretation methods used in paper should be provided for the reimplementation purposes. 

A5. We mentioned the availability of the data and program codes in Data availability statement.

The source codes are freely accessible at https://github.com/kuratahiroyuki/PredIL13. The web application is freely available at http://kurata35.bio.kyutech.ac.jp/PredIL13.

Q6. For the reader concerns the authors are advised to add the recent predictors such as AIPs-SnTCN, DeepAVP-TPPred, Deepstacked-AVPs,iAFPs-Mv-BiTCN and pAVP_PSSMDWT-EnC models 

A6. These predictors are not directly related to prediction of interleukin 13-inducing peptides. Instead of them we added a general review on the computational approach of prediction of anti‑inflammatory in Introduction.

Q7.The feature vector size of the encoding methods should be provided in the paper. 

A7. We described the vector size or the number of descriptors of all the features in the feature encoding method section.

Q8. What are limitations of the proposed model.

A8. We added the Limitation section to Results and discussion. One limitation of this study is that the number of experimentally validated IL-13-inducing peptides is small. Thus, we need to enlarge the dataset to increase the prediction performance. Furthermore, using the large dataset we construct a generative AI to design de novo peptide sequences and to ensure the generated sequences are biologically functional and potentially beneficial for medication.

Reviewer #2:

We revised the title to attract many readers from “PredIL13: stacking a variety of machine and deep learning methods by weight coefficient-based single-feature model selection for identifying IL13-inducing peptides” to “PredIL13: stacking a variety of machine and deep learning methods with ESM-2 language model for identifying IL13-inducing peptides”

Q1. How the authors handle the imbalance data while training the model. Did they use any oversampling or under sampling approach, authors should clearly mention.

A1. We neither used over-sampling nor under-sampling methods. We added this statement in the section of Dataset preparation.

Q2. What is justification of using the machine learning models in the proposed model? As there are several other training models available in the literature.

A2. We focused on widely-used, typical machine and deep learning methods. They would be sufficient to construct the best model or PredIL13.

Q3. As the authors used several feature extraction methods to formulate peptide sequence. However, It should be convincing if the authors highlight the high contributory features using SHAP analysis.

A3. At the section of Importance of each single-feature model, we discussed SHapley Additive exPlanations (SHAP) analysis, while making Figure S2. Feature selection methods other than AWCLR, which represents the contribution of each single-feature model (feature) to the log-odds of binary classification, are known. For example, SHAP analysis provides a more nuanced view by showing how each feature contributes to individual predictions (Figure S2). In this study we used the AWCLR as it has a theoretical intelligible basis to overcome the-state-of-the-art methods. SHAP analysis will be considered in a next step.

Q4. What should be the future direction of the proposed model.

A4. We added the Limitation section to Results and discussion to mention the future direction. One limitation of this study is that the number of experimentally validated IL-13-inducing peptides is small. Thus, we need to enlarge the dataset to increase the prediction performance. Furthermore, using the large dataset we construct a generative AI to design de novo peptide sequences and to ensure the generated sequences are biologically functional and potentially beneficial for medication.

---

## [Decision Letter · Decision Letter 1]

6 Aug 2024

PredIL13: stacking a variety of machine and deep learning methods with ESM-2 language model for identifying IL13-inducing peptides

PONE-D-24-20864R1

Dear Dr. Kurata,

We’re pleased to inform you that your manuscript has been judged scientifically suitable for publication and will be formally accepted for publication once it meets all outstanding technical requirements.

Kind regards,

Shahid Akbar, PhD

Academic Editor

PLOS ONE

Additional Editor Comments (optional):

Reviewers' comments:

Reviewer's Responses to Questions

**Comments to the Author**

1. If the authors have adequately addressed your comments raised in a previous round of review and you feel that this manuscript is now acceptable for publication, you may indicate that here to bypass the “Comments to the Author” section, enter your conflict of interest statement in the “Confidential to Editor” section, and submit your "Accept" recommendation.

Reviewer #1: All comments have been addressed

Reviewer #2: All comments have been addressed

2. Is the manuscript technically sound, and do the data support the conclusions?

Reviewer #1: Yes

Reviewer #2: Yes

3. Has the statistical analysis been performed appropriately and rigorously? 

Reviewer #1: Yes

Reviewer #2: Yes

4. Have the authors made all data underlying the findings in their manuscript fully available?

Reviewer #1: Yes

Reviewer #2: Yes

5. Is the manuscript presented in an intelligible fashion and written in standard English?

Reviewer #1: Yes

Reviewer #2: Yes

6. Review Comments to the Author

Reviewer #1: The required comments are successfully incorporated and I think the paper is significantly improved. The paper is acceptable from my side

Reviewer #2: the authors successfully addressed my previous concerns and now the paper is significantly improved. i think the paper can be accepted . no further comments from my side

7. PLOS authors have the option to publish the peer review history of their article (what does this mean?). If published, this will include your full peer review and any attached files.

Reviewer #1: No

Reviewer #2: No

---

## [Editor Report · Acceptance letter]

13 Aug 2024

PONE-D-24-20864R1 

PLOS ONE

Dear Dr. Kurata, 

I'm pleased to inform you that your manuscript has been deemed suitable for publication in PLOS ONE. Congratulations! Your manuscript is now being handed over to our production team.

Kind regards, 

on behalf of

Dr. Shahid Akbar 

Academic Editor

PLOS ONE